# Evaluating the risk of sleep disorders in subjects with a prior COVID-19 infection

**Jaewhan Kim**[1]*, **Kenechukwu C. Ben-Umeh**[2], **Rachel Weir**[3], **Karen Manotas**[3], **Kristi Kleinschmit**[3], **Aaron Fischer**[4], **Peter Weir**[5], **Fernando Wilson**[6]

**1** Department of Physical Therapy, University of Utah, Salt Lake City, Utah, United States of America, **2** Department of Pharmacotherapy, University of Utah College of Pharmacy, Salt Lake City, Utah, United States of America, **3** Department of Psychiatry, University of Utah, Salt Lake City, Utah, United States of America, **4** Department of Educational Psychology, University of Utah, Salt Lake City, Utah, United States of America, **5** University of Utah Medical Group Population Health, University of Utah, Salt Lake City, Utah, United States of America, **6** Department of Population Health Sciences, University of Utah, Salt Lake City, Utah, United States of America

* Jaewhan.kim@utah.edu

## Abstract

Previous studies have reported a potential occurrence of sleep disorders in patients following a COVID-19 infection. However, these findings were based on surveys or retrospective studies with small sample sizes. This study examined if subjects with a previous COVID-19 infection in 2020 experienced sleep disorders in 2021. Using the 2019–2021 Utah All Payers Claims Database (APCD), adults ($\geq$18 to 62 years old in 2019) covered by private insurance and Medicaid were identified. Sleep disorders were identified from the primary and secondary diagnosis in 2021. Baseline characteristics of subjects such as age, gender, race/ethnicity, type of insurance, and comorbid conditions were identified from the database. Entropy balancing was used to balance the baseline characteristics of subjects with and without a COVID-19 infection in 2020. Weighted logistic regression was used to identify significant factors that were associated with sleep disorders. A total of 413,958 subjects were included in the study. The average (SD) age was 38 (17) years old in 2019 and 58% were female. Among the subjects, about 39% had a COVID-19 infection in 2020. Those who had a COVID-19 infection in 2020 were 53% more likely to have a sleep disorder in 2021 (OR = 1.53; 95% Confidence Interval: 1.48–1.58). Sleep disorders could be one of long-term COVID-19 symptoms. More screening and observations for those who had a COVID-19 infection could be important to improve sleep related problems.

## Introduction

Sleep disorders are a category of health concerns that impede the ability to sleep well without interruptions consistently over a span of time [1]. It is estimated that about 15–25% of the United States (US) population experience different sleep disorders and the most common manifestations include sleep apnea, insomnia, narcolepsy and restless leg syndrome [2–4]. Sleep disorders may be a result of various factors ranging from stressful life events to predisposing factors or medical disorders and are a growing public health concern in the United

**Data Availability Statement:** Data cannot be shared publicly due to regulations set by the Utah Department of Health and Human Services, and the Utah Resource for Genetic and Epidemiologic Research (RGE). Data are available upon approvals

from the University of Utah Institutional Review Board (irb@utah.edu) and the Utah Resource for Genetic and Epidemiologic Research (rge@hsc.utah.edu).

**Funding:** The author(s) received no specific funding for this work.

**Competing interests:** The authors have declared that no competing interests exist.

States [3, 5]. Occurrence of sleep disorders have been tied to several mental health disorders such as major depressive disorder, anxiety, post-traumatic stress disorder, as well as generalized stress [6].

The emergence of the novel coronavirus disease (COVID-19) which eventually morphed into a global pandemic impacted the world in many ways. The lockdown that ensued, the social isolation coupled with fear, and looming uncertainties and economic difficulties led to psychological effects and sleep disorders [7, 8]. There is growing concern that the effect of COVID-19 on the sleep cycle may not just be short term but also long term, which may be due to prolonged disruption of the circadian rhythm [9, 10]. Studies have shown sleep disorders to be a significant neuropsychiatric symptom of post-COVID-19 syndrome [11–13]. A study reported a potential association of sleep disturbances in patients 3–6 months after recovery from COVID-19 [12]. Despite the potential impact of COVID-19 infection and sleep disorder, most previous studies have explored sleep disorders with small sample sizes during the COVID-19 pandemic, but not in the long term after the pandemic with a state-wide large dataset. Hence, the aim of this study was to investigate the relationship between a previous COVID-19 infection and risk of sleep disorders using a population level data from 2019 through 2021.

## Materials and methods

### Data

The 2019–2021 Utah All Payers Claims Database (APCD) was used for the study. The APCD is known to cover over 70% of the Utah population covered by Medicaid, private insurance and Medicare Advantage [14, 15]. The APCD includes medical, pharmacy and dental claims of subjects. The database consists of two major files for research: enrollment file and claims file. The enrollment file provides insurance coverage start/end date, type of insurance (Medicaid, Medicare and private insurance), date of birth, gender, race/ethnicity, Medicaid/Medicare coverage, and medical/pharmacy/dental coverage indicator. The claims file includes service start/end date, type of claims (medical, pharmacy and dental), Current Procedural Terminology (CPT), International Classification of Diseases-10th Revision (ICD-10) diagnosis code, place of service, bill type, provider taxonomy, reimbursement amounts, national drug code and drug name. The data were accessed for research purposes from August 1, 2022 through September 30, 2023. Authors had no access to information that could identify individual participants during or after data collection. This study received an exemption determination from Institutional Review Board (IRB) at the University of Utah (IRB 00151091).

### Subjects

Adults ($\geq$18 to 62 years old in 2019) who had continuous medical enrollment over 36 months (2019 through 2021) were included. Those who had a COVID-19 diagnosis in 2021 were excluded in the study. In addition, those who had sleep disorder diagnoses in 2019 and/or 2020 were excluded. Subjects who were dual eligible (i.e. Medicaid and Medicare) were excluded from the study as well.

### Outcome

Sleep disorders were identified by using the primary and second diagnosis codes in 2021. The ICD-10 diagnosis codes for the sleep disorders (F51.XX, G47.XX, R063) were obtained from Agency for Healthcare Research and Quality (AHRQ) Clinical Classifications Software Refined (CCSR) [16].

## Covariates

COVID-19 diagnosis (Yes/No) in 2020 was the independent variable in the study. The following codes were used to identify the COVID-19 infection (ICD-10 diagnosis codes: J1281, J1282, U071, U099, B948, B9729, Z8616, O985, Z20828; CPT: 86413, 86328, 86769, 87426, 87428, 87635, 87636, 87637, 87811, 87913, C9803) [16, 17]. Demographic information such as age in 2019 (18–30, 31–40, 41–50, and 51–62 years old), gender (male/female), race/ethnicity (non-Hispanic White, non-Hispanic African American, non-Hispanic Pacific Islander/Asian, Hispanic, Unknown), and Medicaid coverage (Yes/No) were included in the regression model. Individual physical and mental comorbid conditions were identified from 2019 using the diagnosis codes from Centers for Medicare and Medicaid (CMS) Chronic Conditions Warehouse and AHRQ CCSR [16, 18]. The comorbid conditions included depression, anxiety, cognitive disorder, personality disorder, schizophrenia, bipolar, eating disorder, attention-deficit/hyperactivity disorder (ADHD), diabetes, heart failure, cerebrovascular disease, opioid use disorder, thyroid disorder, obesity and Chronic Obstructive Pulmonary Disease (COPD)/asthma. These conditions were included because they were known to cause sleep difficulties [19–21].

## Statistical approach

Mean, standard deviation, counts, and percentages were used to summarize the baseline characteristics of the subjects in 2019. To balance the different baseline characteristics of the two groups (i.e. with and without COVID-19 infection), Entropy Balancing (EB) with mean, variance and skewness was used to create weights for the subjects [22, 23]. The EB approach for a binary outcome is based on a maximum entropy reweighting scheme that assigns weights to each subject so that the control group (i.e. no COVID infection group in this study) is reweighted to match the controlled covariate moments (i.e. mean, standard deviation, and variance in this study) in the treatment group (i.e. COVID infection group in this study). Therefore, EB ensures that the two groups being compared are similar enough by reweighting the patient characteristics (i.e. covariates) [23–25]. To calculate a weight for each subject, all of the covariates that were listed above were included. Standardized differences in the variables were calculated to evaluate any differences in the variables between the two groups. Standardized differences less than 0.1 indicated no significant differences in the variables between the two groups. The standardized differences of all of the variables after EB was 0.00, indicating that the baseline variables were balanced well between the two groups. Following EB, T-tests for continuous variables and Chi-square tests for categorical variables were used to compare the baseline characteristics of the subjects in Table 1. Weighted logistic regression was used to identify which baseline variables were associated with sleep disorders in 2021. The weighted regression model controlled for the covariates that were included in the EB [26, 27]. E-value was used to estimate odds ratio that would invalidate the association between the COVID-19 infection and sleep disorder due to unmeasured confounding [28, 29].

Another analysis was performed, categorizing COVID-19 infection severity into three groups: no infection, mild COVID-19 infection, and severe COVID-19 infection. Subjects with any emergency room (ER) visits or hospital admissions within 7 days before or after a COVID-19 infection diagnosis were categorized as having a severe COVID-19 infection, while those without such visits or hospital admissions were categorized as having a mild COVID-19 infection. The area under the receiver operating characteristic (ROC) curve (AUC) was used to measure the performance of the logistic regression models [30]. P-value less than 0.05 was defined as statistically significant. Stata version 18.0 was used for the analysis.

**Table 1. Summary statistics of subjects in 2019 after entropy balancing.**

| Variable | Overall (n = 413,958; 100%) | COVID infection in 2020 | | p-value |
|---|---|---|---|---|
| | | No (n = 251,022; 60.64%) | Yes (n = 162,936; 39.36%) | |
| | mean (SD)/ N(%) | mean (SD)/N(%) | mean (SD)/N(%) | |
| Age (as continuous) | 38.44 (12.90) | 38.44 (14.28) | 38.44 (11.38) | 0.96 |
| Age category | | | | 0.96 |
| Age 18 to 30 | 130,438 (31.51%) | 79,097 (31.51%) | 51,341 (31.51%) | |
| Age 31 to 40 | 102,579 (24.78%) | 62,203 (24.78%) | 40,376 (24.78%) | |
| Age 41 to 50 | 91,609 (22.13%) | 55,551 (22.13%) | 36,058 (22.13%) | |
| Age 51 to 62 | 89,332 (21.58%) | 54,171 (21.58%) | 35,162 (21.58%) | |
| Female | 240,054 (57.99%) | 145,568 (57.99%) | 94,487 (57.99%) | 0.89 |
| Race/Ethnicity | | | | 0.97 |
| Non-Hispanic White | 60,686 (14.66%) | 36,825 (14.67%) | 23,870 (14.65%) | |
| Non-Hispanic Black | 1,490 (0.36%) | 904 (0.36%) | 587 (0.36%) | |
| Non-Hispanic Asian/Pacific Islander/American Indian | 3,933 (0.95%) | 2,385 (0.95%) | 1,548 (0.95%) | |
| Hispanic | 9,935 (2.40%) | 6,025 (2.40%) | 3,910 (2.40%) | |
| Unknown | 337,914 (81.63%) | 204,884 (81.62%) | 133,021 (81.64%) | |
| Medicaid coverage | 34,441 (8.32%) | 20,910 (8.33%) | 13,540 (8.31%) | 0.82 |
| Comorbid condition | | | | |
| Depression | 75,879 (18.33%) | 46,012 (18.33%) | 29,882 (18.34%) | 0.96 |
| Anxiety | 62,466 (15.09%) | 37,879 (15.09%) | 24,587 (15.09%) | 0.97 |
| Cognitive disorder | 14,861 (3.59%) | 9,012 (3.59%) | 5,849 (3.59%) | 0.98 |
| Personality disorder | 1,242 (0.30%) | 753 (0.30%) | 489 (0.30%) | 1.00 |
| Schizophrenia | 2,194 (0.53%) | 1,330 (0.53%) | 864 (0.53%) | 0.92 |
| Bipolar | 7,037 (1.70%) | 4,267 (1.70%) | 2,770 (1.70%) | 1.00 |
| Eating disorder | 911 (0.22%) | 552 (0.22%) | 358 (0.22%) | 1.00 |
| ADHD | 13,371 (3.23%) | 8,108 (3.23%) | 5,263 (3.23%) | 0.99 |
| Diabetes | 21,898 (5.29%) | 13,279 (5.29%) | 8,619 (5.29%) | 0.99 |
| Heart failure | 952 (0.23%) | 577 (0.23%) | 375 (0.23%) | 1.00 |
| Cerebrovascular disease | 1,780 (0.43%) | 1,079 (0.43%) | 701 (0.43%) | 1.00 |
| Opioid use disorder | 4,802 (1.16%) | 2,912 (1.16%) | 1,890 (1.16%) | 1.00 |
| Thyroid disorder | 22,933 (5.54%) | 13,907 (5.54%) | 9,043 (5.55%) | 0.97 |
| Obesity | 70,456 (17.02%) | 42,724 (17.02%) | 27,732 (17.02%) | 0.96 |
| Asthma/COPD | 11,384 (2.75%) | 6,903 (2.75%) | 4,481 (2.75%) | 0.99 |
| Outcome | | | | |
| Sleep disorders | 15,689 (3.79%) | 6,903 (2.94%) | 8,293 (5.09%) | <0.01 |

Abbreviations: *SD*, Standard deviation; *COPD*, Chronic obstructive pulmonary disease; *ADHD*, Attention-deficit hyperactivity disorder.

## Results

A total of 1,069,177 subjects were continuously covered by insurance for 36 months. Among them, only 564,045 subjects met the age criteria ($\geq$18 to 62 years old in 2019). Those who had a sleep disorder diagnosis in 2019 and 2020 (n = 34,994) or were dual eligible at any time (n = 26,487) were excluded. In addition, those with COVID-19 infection in 2021 (n = 88,606) were excluded. The total number of the subjects in the analysis was 413,958. Among the study subject, 39.36% (n = 162,936) had a COVID-19 infection in 2020 (Fig 1).

A total of 2.94% of the subjects who had no COVID-19 infection in 2020 and 5.09% of subjects who had COVID-19 infection had a sleep disorder that was newly diagnosed in 2021

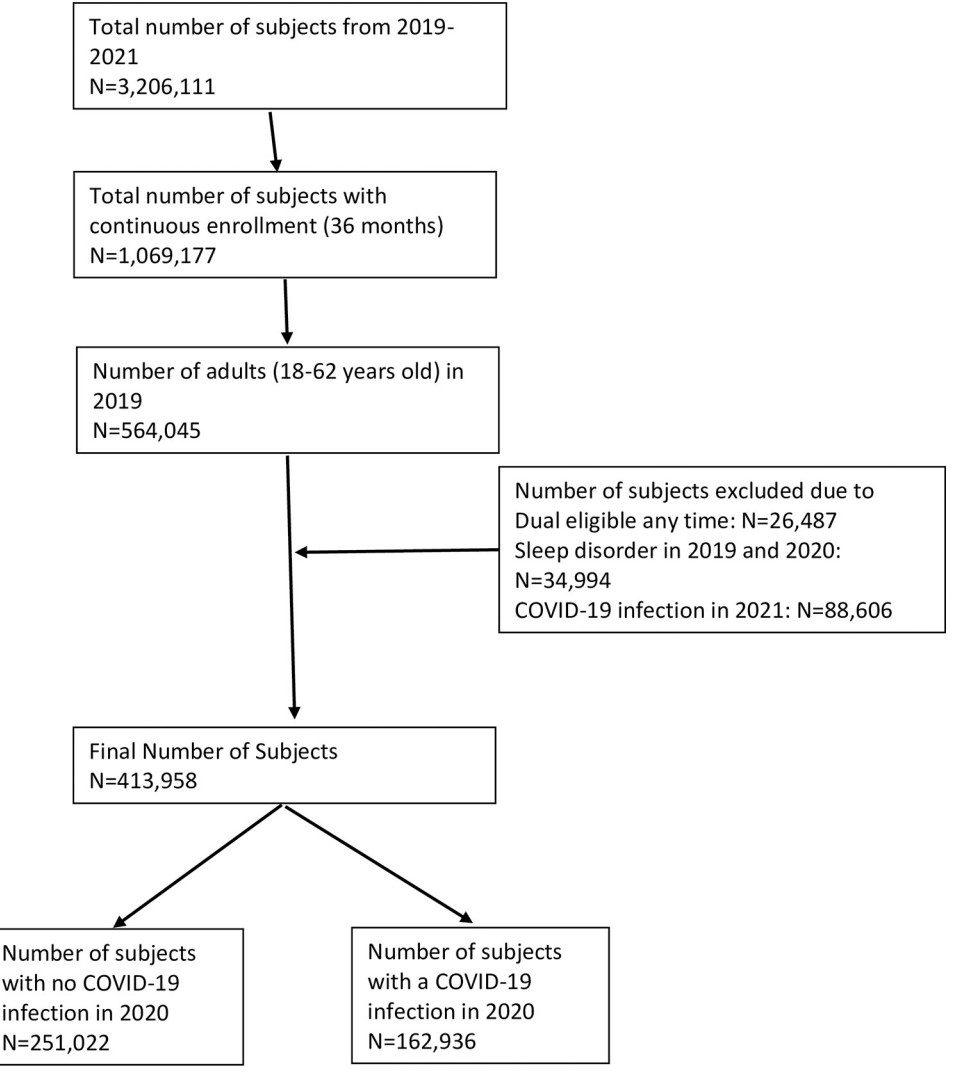

**Fig 1. Flowchart for the cohort selection.**

(p<0.01). Average (SD) age was 38 (17) years old in 2019, and about 58% were female. About 18% had a depression diagnosis in 2019 and 17% had an obesity diagnosis in 2019 (Table 1).

Those who had a COVID-19 infection in 2020 were 53% more likely to have a sleep disorder in 2021 (OR = 1.53, p<0.01). As subjects got older, they were more likely to experience a sleep disorder by 39% (OR = 1.39, p<0.01 for 31–40 years old), by 81% (OR = 1.84, p<0.01 for 41–50 years old), and 107% (OR = 2.07, p<0.01 for 51–62 years old) as compared to the younger age group (18–30 years old in 2019). Females were less likely to have a sleep disorder than males by 18% (OR = 0.82, p<0.01). Subjects who had mental illnesses such as depression (OR = 1.73, p<0.01), anxiety (OR = 1.59, p<0.01), eating disorders (OR = 1.43, p = 0.02), ADHD (OR = 1.20, p<0.01) and cognitive disorder (OR = 1.43, p<0.01) were found to have a higher likelihood of experiencing a sleep disorder. Physical conditions such as heart failure (OR = 1.74, p<0.01), opioid use disorder (OR = 1.39, p<0.01), thyroid disorder (OR = 1.16, p<0.01), and obesity (OR = 2.84, p<0.01) were significantly associated with a higher risk of sleep disorder in 2021 (Table 2).

**Table 2. Weighted logistic regression of association between a prior COVID-19 infection and risk of sleep disorders in all subjects in 2019 (n = 413,958).**

| Covariate | Odds ratio | p-value | 95% confidence interval | |
|---|---|---|---|---|
| COVID infection | 1.53 | <0.01 | 1.48 | 1.58 |
| Age | | | | |
| Age 18 to 30 | reference | | | |
| Age 31 to 40 | 1.39 | <0.01 | 1.32 | 1.46 |
| Age 41 to 50 | 1.84 | <0.01 | 1.75 | 1.93 |
| Age 51 to 62 | 2.07 | <0.01 | 1.97 | 2.18 |
| Female | 0.82 | <0.01 | 0.80 | 0.85 |
| Race/Ethnicity | | | | |
| Non-Hispanic White | reference | | | |
| Non-Hispanic Black | 0.78 | 0.08 | 0.60 | 1.03 |
| Non-Hispanic Asian/Pacific Islander/American Indian | 0.77 | <0.01 | 0.64 | 0.92 |
| Hispanic | 0.81 | <0.01 | 0.72 | 0.92 |
| Unknown | 0.86 | <0.01 | 0.82 | 0.90 |
| Medicaid coverage | 0.94 | 0.08 | 0.88 | 1.01 |
| Depression | 1.73 | <0.01 | 1.65 | 1.81 |
| Anxiety | 1.59 | <0.01 | 1.52 | 1.67 |
| Cognitive disorder | 1.43 | <0.01 | 1.28 | 1.-60 |
| Personality disorder | 1.34 | 0.03 | 1.03 | 1.74 |
| Schizophrenia | 0.98 | 0.79 | 0.82 | 1.16 |
| Bipolar | 1.05 | 0.36 | 0.95 | 1.17 |
| Eating disorder | 1.43 | 0.02 | 1.07 | 1.91 |
| ADHD | 1.20 | <0.01 | 1.07 | 1.36 |
| Diabetes | 0.93 | 0.04 | 0.87 | 0.99 |
| Heart failure | 1.74 | <0.01 | 1.39 | 2.17 |
| Cerebrovascular disease | 1.16 | 0.16 | 0.94 | 1.42 |
| Opioid use disorder | 1.39 | <0.01 | 1.23 | 1.58 |
| Thyroid disorder | 1.16 | <0.01 | 1.08 | 1.24 |
| Obesity | 2.84 | <0.01 | 2.73 | 2.95 |
| Asthma/COPD | 1.33 | <0.01 | 1.22 | 1.46 |

Abbreviation: *COPD*, Chronic obstructive pulmonary disease; *ADHD*, Attention-deficit hyperactivity disorder.

E-value was 2.43, which was greater than any odds ratio of the COVID-19 infection variable in the regression. Because this odds ratio was bigger than any other odds ratios of the controlled variables in the regression, it may indicate that the regression model might not be impacted by unobservable confounders. The AUC following the logistic regressions were 0.72 for both Tables 2 and 3, indicating that there is a 72% chance of the models distinguishing between positive and negative cases.

About 61% (n = 251,022) of subjects had no COVID-19 infection, 13% (n = 55,165) had mild COVID-19 infection, and 26% (n = 107,771) had severe COVID-19 infection. The analysis of COVID-19 infection severity categories showed that subjects with severe COVID-19 infection had a higher risk of experiencing sleep disorders than those with mild or no COVID-19 infection. Compared to subjects with no COVID-19 infection, those with mild COVID-19 infection were 25% more likely to have sleep disorders in the following year (OR = 1.25, p<0.01), while subjects with severe COVID-19 infection had a 64% higher likelihood of having sleep disorders (OR = 1.64, p<0.01). To compare those with mild infection to those with severe infection, we conducted a Wald test resulting in a statistically significant difference (F-

**Table 3. Association between COVID-19 severity and risk of sleep disorders.**

| Covariate | Odds ratio | p-value | 95% confidence interval | |
|---|---|---|---|---|
| Severity of COVID-19 | | | | |
| No COVID infection | Reference | | | |
| Mild COVID | 1.25 | <0.01 | 1.19 | 1.32 |
| Severe COVID | 1.64 | <0.01 | 1.58 | 1.70 |
| Age | | | | |
| Age 18 to 30 | Reference | | | |
| Age 31 to 40 | 1.39 | <0.01 | 1.32 | 1.46 |
| Age 41 to 50 | 1.82 | <0.01 | 1.73 | 1.91 |
| Age 51 to 62 | 2.03 | <0.01 | 1.93 | 2.14 |
| Female | 0.81 | <0.01 | 0.78 | 0.84 |
| Race/Ethnicity | | | | |
| Non-Hispanic White | Reference | | | |
| Non-Hispanic Black | 0.78 | 0.07 | 0.59 | 1.02 |
| Non-Hispanic Asian/Pacific Islander/American Indian | 0.76 | <0.01 | 0.64 | 0.92 |
| Hispanic | 0.81 | <0.01 | 0.72 | 0.92 |
| Unknown | 0.88 | <0.01 | 0.84 | 0.92 |
| Medicaid coverage | 0.94 | 0.06 | 0.88 | 1.00 |
| Depression | 1.72 | <0.01 | 1.65 | 1.80 |
| Anxiety | 1.59 | <0.01 | 1.52 | 1.67 |
| Cognitive disorder | 1.43 | <0.01 | 1.27 | 1.59 |
| Personality disorder | 1.34 | 0.03 | 1.03 | 1.73 |
| Schizophrenia | 0.97 | 0.72 | 0.82 | 1.15 |
| Bipolar | 1.05 | 0.39 | 0.94 | 1.16 |
| Eating disorder | 1.42 | 0.02 | 1.06 | 1.90 |
| ADHD | 1.21 | <0.01 | 1.07 | 1.36 |
| Diabetes | 0.93 | 0.02 | 0.87 | 0.99 |
| Heart failure | 1.72 | <0.01 | 1.38 | 2.15 |
| Cerebrovascular disease | 1.14 | 0.19 | 0.93 | 1.40 |
| Opioid use disorder | 1.38 | <0.01 | 1.22 | 1.56 |
| Thyroid disorder | 1.15 | <0.01 | 1.08 | 1.23 |
| Obesity | 2.82 | <0.01 | 2.71 | 2.93 |
| Asthma/COPD | 1.33 | <0.01 | 1.22 | 1.45 |

value = 348, p<0.01) between the two groups. This indicated that those with severe infection experienced higher rates of sleep disorders than those with mild infection (Table 3).

## Discussion

Our study utilizing the Utah All Payers Claims Database (APCD) showed that those who had COVID-19 infection in 2020 had over 50% higher chances of experiencing a sleep disorder in 2021 compared patients who did not have COVID-19 infection in 2020. Despite the widespread impact of the COVID-19 pandemic and the possibility of subsequent development of sleep disorders among millions, few previous studies have used robust population-level data to assess the long-term likelihood of developing sleep disorders following a COVID-19 infection. We use 3 years of comprehensive longitudinal data that capture patients covered by commercial insurance, Medicaid and Medicare advantage, enabling an in-depth analysis.

In a systematic review and meta-analysis that included 177 studies with a total of 345,270 participants across 39 countries, Alimoradi et al estimated the prevalence of sleep disorders during the COVID-19 pandemic and its relationship with psychological distress [30]. The study reported the prevalence of sleep disorders to be 57% among COVID-19 patients and 18% in the general population and the sleep problems were attributed to psychological distress such as depression and anxiety [31]. Similarly, Tasnim et al also reported an increased rate of sleep disorders during the pandemic [7]. Moreover, more studies have reported sleep disorders, among many other symptoms, as long-term effects associated with a prior COVID-19 infection [32–35]. This has been referred to as post-COVID-19 syndrome or long-COVID. Our results are consistent with the findings from these studies.

Premraj et al in a meta-analysis of over 10,000 patients in 18 studies reported a 31% prevalence of sleep disturbances as a neuropsychiatric post-COVID-19 symptom [12]. Surprisingly, these neuropsychiatric post-COVID-19 symptoms were significantly higher when accessed long-term (6 or more months after an infection) compared to mid-term (3 to 6 months) which may indicate that the symptoms are more likely to develop and not persist over time post-infection [12]. This is contrary to the results from another retrospective study that reported higher rates of sleep disorders at 6 months after an acute COVID-19 diagnosis compared to 2 years, showing declining post-COVID symptoms [33].

A cohort study that explored the long-term health consequences among patients who had been hospitalized for COVID-19 found that such patients reported having sleep disorders at a higher rate (26%) 6 months after an acute COVID-19 infection [31]. The severity of illness as well as being a woman were important risk factors for the long-term effects identified in this study [36]. Our study, however, shows being female as associated with a lower likelihood of having a sleep disorder in 2021. The study included only patients who had been discharged after hospitalization, hence it was not clear if the observed effect would have been different for patients who were not hospitalized. On one hand, Taquet et al in a retrospective study that estimated the incidence rates of neurological and psychiatric diagnoses in 236,379 patients 6 months after COVID-19 diagnosis reported lower rates of insomnia in patients who were not hospitalized when they had COVID-19 infection compared to patients who were hospitalized [32]. On the other hand, this was inconsistent with findings from another study that reported higher rates of sleep disorders among those non-hospitalized compared to those hospitalized when they had COVID-19 infection [12]. Patients with more severe infections were also found to have higher rates of both neurological and psychiatric outcomes months after diagnosis [32, 37].

The mechanism of the association between COVID-19 infection and sleep disorders is thought to be multifactorial and could encompass factors such as social isolation, ICU stay, direct effect of the viral infection, immunological response, cerebrovascular changes, medication use, among many others [37, 38]. In addition to infecting peripheral tissues linked to the central nervous system (CNS), the SARS-CoV-2 coronavirus also directly enters the brain by permeating the blood-brain barrier (BBB) [39, 40]. This neuroinvasion could potentially affect brain functions, including sleep regulation [40]. Furthermore, hyperinflammatory reactions where the concentrations of C-reactive proteins (CRP) and interleukin-6 (IL-6) are increased may also be implicated in the association between COVID-19 infection and psychiatric symptoms [41]. Specifically, there is a growing body of evidence that COVID-19 infection can lead to elevated levels of inflammatory mediators such as cytokines [42]. Certain cytokines, including IL-6, IL-1α, and tumor necrosis factor (TNF-α), exhibit notable circadian oscillations when they enter the brain, leading to sleep disorders [43, 44]. This relationship between inflammatory mediators and psychiatric symptoms has been previously demonstrated through the link between inflammation and depression [45]. Social isolation also plays a key role in

sleep disorders. Sleep disorders were reported to increase during quarantine periods due to COVID-19 infection [46]. During COVID-19 lockdowns, insomnia, poor sleep maintenance, and reduced sleep quality increased for the general population in several countries. This has been attributed to the effect of confinement, worsened by the psychological effects of the disease and increased exposure to artificial lighting from electronic devices [6, 47, 48].

However, not all studies showed a positive association between COVID-19 infection and sleep disorders. A meta-analysis that investigated the evidence of long-term post-COVID symptoms among children and young people reported non-significant pooled risk difference in post-COVID cases compared to controls for insomnia [49]. It is particularly noteworthy that about half of the studies included in the meta-analysis had a high risk of bias and high heterogeneity which was a limitation.

Sleep disorders are recognized as important public health and economic concerns requiring prompt attention [50]. Thus, our study's findings have various implications for healthcare providers and policymakers. This study contributes to the existing body of literature available to healthcare providers, highlighting the potential impact of COVID-19 infection on sleep. It may prompt providers to consider incorporating sleep-related questions into patient assessments, particularly considering that symptoms of sleep disorders may manifest in different ways and could be difficult to link specifically to the infection, especially in the absence of definitive diagnostics [51]. Additionally, healthcare providers can offer psychological support to patients who may feel unheard when complaining about their symptoms [51]. Healthcare providers could be instrumental in managing sleep disturbances, as well as in assessing patients' progress through sleep pattern tracking and assisting patients in setting realistic goals for their recovery [52]. Cognitive behavioral therapy and progressive muscle relaxation are interventions that have been used to treat sleep disorders in patients with COVID-19 infection [53, 54]. In a randomized controlled trial, Liu et al reported that PMR reduced anxiety levels and improved sleep quality after 5 days [54]. The findings of this study were inconsistent with that of another study by Masih et al which showed no difference between intervention groups [55]. Other alternative therapies include short-term benzodiazepines and hypnosis [56].

Moreover, our study generates relevant scientific evidence that will better equip policymakers in their decision-making process [57]. Policies that could stem from our findings encompass launching public health campaigns aimed at raising awareness about the potential impact of COVID-19 on sleep and the importance of seeking timely medical attention for sleep-related issues. Other policies include establishing integrated care models that would ensure collaborative care and allocating funding for research and surveillance to sustain continued research into the relationship between COVID-19 and sleep disorders.

Even though our study used a state-wide dataset to identify COVID-19 infection and sleep disorders, there were some limitations to consider. First, some subjects who might have had COVID-19 infection might not have been seen by doctors. Thus, there could be an underreporting issue. Also, it is known that there were subjects who had COVID-19 infection with no symptoms. Second, the results of the study may not be generalizable to other states as the data used is specific to the state of Utah in the United States. In addition, these findings could not be generalizable to adolescents or the elderly population. Third, prescription medications such as antidepressants and medications for pain, hypertension and asthma that could interfere with sleep were not considered in the study. On the other hand, some antidepressant medications could be prescribed for sleep disorders, but these were not considered to identify patients with sleep disorder. Continuous positive airway pressure (CPAP) therapy is used for patients who have obstructive sleep apnea, but this therapy was not considered to identify sleep disorder in this study. Fourth, the reliability of the race/ethnicity variables may be compromised by the significant amount of missing information in the category. Therefore, the estimates

obtained for race/ethnicity may be unreliable. Finally, while we controlled for observed variables that were potentially associated with the cause of sleep disorders, unmeasured confounders could potentially influence the results of the study, given the observational study design.

## Conclusions

Utilizing a population database, this study has unveiled a potential correlation between COVID-19 infection and an increased likelihood of experiencing sleep disorders. Further investigations are essential to elucidate the potential mechanism underlying the connection between COVID-19 and sleep disturbances.

## Supporting information

**S1 Table. Standardized differences before and after entropy balancing.** Abbreviation: *COPD*, Chronic obstructive pulmonary disease; *ADHD*, Attention-deficit hyperactivity disorder.
(DOCX)

## Author Contributions

**Conceptualization:** Jaewhan Kim, Kenechukwu C. Ben-Umeh, Rachel Weir, Karen Manotas, Kristi Kleinschmit, Aaron Fischer, Peter Weir, Fernando Wilson.

**Data curation:** Jaewhan Kim.

**Formal analysis:** Jaewhan Kim.

**Funding acquisition:** Fernando Wilson.

**Investigation:** Jaewhan Kim, Kenechukwu C. Ben-Umeh, Rachel Weir, Karen Manotas, Kristi Kleinschmit, Aaron Fischer, Peter Weir.

**Methodology:** Jaewhan Kim.

**Resources:** Fernando Wilson.

**Supervision:** Jaewhan Kim.

**Validation:** Jaewhan Kim, Rachel Weir, Karen Manotas, Kristi Kleinschmit, Aaron Fischer, Peter Weir, Fernando Wilson.

**Writing – original draft:** Jaewhan Kim, Kenechukwu C. Ben-Umeh, Rachel Weir, Karen Manotas, Kristi Kleinschmit, Aaron Fischer, Peter Weir, Fernando Wilson.

**Writing – review & editing:** Jaewhan Kim, Kenechukwu C. Ben-Umeh, Rachel Weir, Karen Manotas, Kristi Kleinschmit, Aaron Fischer, Peter Weir, Fernando Wilson.

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
