## [Decision Letter · Decision Letter 0]

4 Mar 2024

PONE-D-23-38335Evaluating the risk of sleep disorders in subjects with a prior COVID-19 infectionPLOS ONE

Dear Dr. Kim,

Thank you for submitting your manuscript to PLOS ONE. After careful consideration, we feel that it has merit but does not fully meet PLOS ONE’s publication criteria as it currently stands. Therefore, we invite you to submit a revised version of the manuscript that addresses the points raised during the review process.

We look forward to receiving your revised manuscript.

Kind regards,

Braja Gopal Patra, Ph.D.

Academic Editor

PLOS ONE

Journal Requirements:

Additional Editor Comments:

This manuscript discusses sleep disorder prior to COVID-19 infection in the UTAH all payers claim database. It is an interesting study, however, there exist several concerns.

1. It will be great to have patient counts beside the patient percentages in tables and tests.

2. In Table 1, most of the % of the patients are same or different by 0.01% except for sleep disorder. Why is that?

3. There is a mention of 101% for 51-62 years old, is this a typo?

4. It will be great to have a flowchart for cohort selection.

5. In general, table 1 is very confusing. The values in the table look repetitive and it is not outlined how those values have been calculated. Apparently p values have been used to distinguish both categorical and continuous data, but it is not clear what statistical test has been used. Further, entropy balancing has not been cited.

6. Weighted logistic regression meaning ridge/lasso/something else?

7. Line 125: P-value written as E-value

Reviewers' comments:

Reviewer's Responses to Questions

**Comments to the Author**

1. Is the manuscript technically sound, and do the data support the conclusions?

Reviewer #1: Yes

Reviewer #2: Partly

2. Has the statistical analysis been performed appropriately and rigorously? 

Reviewer #1: Yes

Reviewer #2: No

3. Have the authors made all data underlying the findings in their manuscript fully available?

Reviewer #1: Yes

Reviewer #2: No

4. Is the manuscript presented in an intelligible fashion and written in standard English?

Reviewer #1: Yes

Reviewer #2: No

5. Review Comments to the Author

Reviewer #1: Your paper "Evaluating the Risk of Sleep Disorders in Subjects with a Prior COVID-19 Infection" offers a noteworthy exploration into the long-term effects of COVID-19 on sleep health. The use of the Utah All Payers Claims Database gives your study a solid foundation, allowing for a comprehensive analysis of post-COVID-19 sleep disorders. Your choice of statistical methods, particularly weighted logistic regression and entropy balancing, effectively addresses potential confounding factors, strengthening the validity of your findings.

However, I would like to point out a couple of areas for potential improvement. The study's geographical focus on Utah might limit the generalizability of your results to broader populations. Additionally, the exclusion of non-prescription medication use and undiagnosed COVID-19 cases could impact the overall picture of sleep disorder prevalence in the post-COVID context.

Despite these limitations, your research is an important contribution to understanding the aftermath of COVID-19. It paves the way for further studies, which I hope will expand on your work by incorporating a more diverse set of data and examining additional factors that might influence sleep health after COVID-19 infection. Your work is crucial in highlighting the need for ongoing research into the long-term health consequences of this global pandemic.

Reviewer #2: 1. A careful proofread is recommended to correct any minor typographical errors.

2. The authors should ensure data availability in line with PLOS's requirements. If there are restrictions, these should be specified.

3. A more in-depth discussion on the potential pathophysiological mechanisms linking COVID-19 to sleep disorders would enhance the paper's contribution to the existing body of knowledge.

4. The authors should clarify the severity of COVID-19 cases included in the study and discuss how this factor might influence the outcomes.

5. The paper would benefit from a limitations section that discusses the potential for unmeasured confounding, given the observational study design.

6. In the results section, the authors should discuss the practical implications of their findings for healthcare providers and public health policy.

6. PLOS authors have the option to publish the peer review history of their article (what does this mean?). If published, this will include your full peer review and any attached files.

Reviewer #1: No

Reviewer #2: No

---

## [Author Response · Author response to Decision Letter 0]

9 Apr 2024

April 05, 2024

Dear Dr. Patra and Reviewers, 

We appreciate the opportunity to revise our submitted manuscript, Evaluating the risk of sleep disorders in subjects with a prior COVID-19 infection. Comments from the reviewers were very helpful in improving the article's focus and we believe the suggestions have made this paper much stronger. Below are the specific reviewer’s comments (in bold) followed by our responses. Newly added parts in the manuscript are highlighted in yellow. Thank you again for taking the time to consider our manuscript. 

Journal Requirements:

Response: We double-checked the style requirements as well as the title, author, affiliations formatting guidelines. Everything is in good shape.

Response: Data are owned by Utah Department of Health and Human Services, thus requiring approvals from the University of Utah Institutional Review Board and the Utah Resource for Genetic and Epidemiologic Research. We made changes in the Data Availability in the manuscript as follows:

Data cannot be shared publicly due to regulations set by the Utah Department of Health and Human Services, and the Utah Resource for Genetic and Epidemiologic Research (RGE). However, data are available upon approvals from the University of Utah Institutional Review Board (irb@utah.edu) and the Utah Resource for Genetic and Epidemiologic Research (rge@hsc.utah.edu)

Additional Editor Comments:

This manuscript discusses sleep disorder prior to COVID-19 infection in the UTAH all payers claim database. It is an interesting study, however, there exist several concerns.

1. It will be great to have patient counts beside the patient percentages in tables and tests.

Response: Thank you for the comment. We have added patient counts in Table 1 so that patient counts in Table 1 can be informative for Tables 2-4. 

Additionally, in the first sentence under the Statistical Approach, we have added “counts” as follows: 

Mean, standard deviation, counts, and percentages were used to summarize the baseline characteristics of the subjects in 2019.

In Table 1, we have added one more row labeled “Outcome” just above the sleep disorders to clarify that the sleep disorders is the study outcome. 

In Tables 2-4, we have corrected the mistakenly reported p-values of 0.00 to <0.01. 

2. In Table 1, most of the % of the patients are same or different by 0.01% except for sleep disorder. Why is that?

Response: Thank you for the comment. We implemented Entropy Balancing (EB) as a method to balance observed variables between the two groups (i.e. subjects with and without a COVID-19 infection). EB is often used to address confounding in observational studies. After calculating weights for each individual based on EB, statistical insignificance of the variables in Table 1, except for the outcome variable (sleep disorder), indicated that the two groups at baseline (prior to COVID-19 infection) were well balanced. Therefore, potential selection bias was eliminated, and the regression aimed to determine if COVID-19 infection was the primary factor associated with the incidence of a sleep disorder. EB is similar to Inverse Probability Weighting (IPW), but it is known that EB is better than IPW in terms of the accuracy of balancing. 

3. There is a mention of 101% for 51-62 years old, is this a typo?

Response: Thank you for the question. 101% is correct because the odds ratio of this age group (51-62) compared to the reference group (i.e. 18-30 years old). was 2.01 

4. It will be great to have a flowchart for cohort selection.

Response: Thank you for the suggestion. We have added a flowchart for cohort selection as Figure 1 in the revision. 

Figure 1. Flowchart for the cohort selection 

5. In general, table 1 is very confusing. The values in the table look repetitive and it is not outlined how those values have been calculated. Apparently, p values have been used to distinguish both categorical and continuous data, but it is not clear what statistical test has been used. Further, entropy balancing has not been cited.

Response: The repetitive numbers in Table 1 were a result of balancing the observed variables through Entropy Balancing (EB) process. We aimed to demonstrate the elimination of potential selection bias and confounding associated with the observed variables. This crucial information indicates that selection bias and confounding might be mitigated through balancing the observed variables. In the Statistical Approach section, we described the statistical tests used for comparing the variables in Table 1 as follows: 

Following EB, T-tests for continuous variables and Chi-square tests for categorical variables were used to compare the baseline characteristics of the subjects in Table 1.

To provide clarity on EB, we have added two additional references:

Hainmueller J. Entropy Balancing for Causal Effects: A Multivariate Reweighting Method to Produce Balanced Samples in Observational Studies. Political Analysis. 2012;20(1):25-46. doi:10.1093/pan/mpr025

Zhao, Qingyuan and Percival, Daniel. Entropy Balancing is Doubly Robust. Journal of Causal Inference, vol. 5, no. 1, 2017, pp. 20160010. https://doi.org/10.1515/jci-2016-0010

6. Weighted logistic regression meaning ridge/lasso/something else?

Response: The Entropy Balancing (EB) method generated weighting values for individuals in the analysis. These weights aim to balance the two groups across the control variables as presented in Table 1. It is necessary to incorporate these weights in the logistic regression, thereby conducting a weighted logistic regression. 

7. Line 125: P-value written as E-value

Response: Thank you for verifying the information. The E-value is correct. The E-value has been used to assess the magnitude of potential unmeasured confounders for controlled variables, including the primary independent variables. We have added two references for the E-value as follows: 

VanderWeele TJ, Ding P. Sensitivity Analysis in Observational Research: Introducing the E-Value. Ann Intern Med. 2017 Aug 15;167(4):268-274. doi: 10.7326/M16-2607. Epub 2017 Jul 11. PMID: 28693043.

Haneuse S, VanderWeele TJ, Arterburn D. Using the E-Value to Assess the Potential Effect of Unmeasured Confounding in Observational Studies. JAMA. 2019;321(6):602–603. doi:10.1001/jama.2018.21554

Reviewers' comments:

Reviewer's Responses to Questions

Reviewer #1: Your paper "Evaluating the Risk of Sleep Disorders in Subjects with a Prior COVID-19 Infection" offers a noteworthy exploration into the long-term effects of COVID-19 on sleep health. The use of the Utah All Payers Claims Database gives your study a solid foundation, allowing for a comprehensive analysis of post-COVID-19 sleep disorders. Your choice of statistical methods, particularly weighted logistic regression and entropy balancing, effectively addresses potential confounding factors, strengthening the validity of your findings.

However, I would like to point out a couple of areas for potential improvement. The study's geographical focus on Utah might limit the generalizability of your results to broader populations. Additionally, the exclusion of non-prescription medication use and undiagnosed COVID-19 cases could impact the overall picture of sleep disorder prevalence in the post-COVID context. Despite these limitations, your research is an important contribution to understanding the aftermath of COVID-19. It paves the way for further studies, which I hope will expand on your work by incorporating a more diverse set of data and examining additional factors that might influence sleep health after COVID-19 infection. Your work is crucial in highlighting the need for ongoing research into the long-term health consequences of this global pandemic.

Response: Thank you for the comments. We agree with these points. While this manuscript offers several significant contributions, we recognize the importance of considering two limitations. The current data are specific to Utah, potentially limiting the generalizability of our results to other states. In addition, the data do not include non-prescription medications and undiagnosed COVID-19 cases, as they only provide information covered by insurance companies. We already included these limitations in the Discussion section and hope future studies address these limitations. 

Second, the results of the study may not be generalizable to other states as the data used is specific to the state of Utah in the United States. In addition, these findings could not be generalizable to adolescents or the elderly population. Third, prescription medications such as antidepressants and medications for pain, hypertension and asthma that could interfere with sleep were not considered in the study. On the other hand, some antidepressant medications could be prescribed for sleep disorders, but these were not considered to identify patients with sleep disorder. Continuous positive airway pressure (CPAP) therapy is used for patients who have obstructive sleep apnea, but this therapy was not considered to identify sleep disorder in this study.

Reviewer #2: 1. A careful proofread is recommended to correct any minor typographical errors.

Response: Thank you for the comment. All authors went through the manuscript to identify and correct any typographical errors. Any changes made during the review were highlighted in the manuscript. 

2. The authors should ensure data availability in line with PLOS's requirements. If there are restrictions, these should be specified.

Response: Thank you for the comment. Data are owned by Utah Department of Health and Human Services. Thus, approvals from the University of Utah Institutional Review Board and the Utah Resource for Genetic and Epidemiologic Research are required. We made changes in the Data Availability in the manuscript as follows:

Data cannot be shared publicly because of the regulations of Utah Department of Health and Human Services, and the Utah Resource for Genetic and Epidemiologic Research (RGE). Data are available after approvals from the University of Utah Institutional Review Board (irb@utah.edu) and the Utah Resource for Genetic and Epidemiologic Research (rge@hsc.utah.edu)

3. A more in-depth discussion on the potential pathophysiological mechanisms linking COVID-19 to sleep disorders would enhance the paper's contribution to the existing body of knowledge.

Response: Thank you for your comment. We agree with the need to expand this section of the discussion. We have added an in-depth discussion highlighting the potential mechanism linking COVID-19 to sleep disorders. We have also added relevant references. Here is the edited paragraph with the added discussion sentences and references highlighted:

The mechanism of the association between COVID-19 infection and sleep disorders is thought to be multifactorial and could encompass factors such as social isolation, ICU stay, direct effect of the viral infection, immunological response, cerebrovascular changes, medication use, among many others [32, 33]. In addition to infecting peripheral tissues linked to the central nervous system (CNS), the SARS-CoV-2 coronavirus also directly enters the brain by permeating the blood-brain barrier (BBB) [34, 35]. This neuroinvasion could potentially affect brain functions, including sleep regulation [35]. Furthermore, hyperinflammatory reactions where the concentrations of C-reactive proteins (CRP) and interleukin-6 (IL-6) are increased may also be implicated in the association between COVID-19 infection and psychiatric symptoms [36]. Specifically, there is a growing body of evidence that COVID-19 infection can lead to elevated levels of inflammatory mediators such as cytokines [37]. Certain cytokines, including IL-6, IL-1α, and tumor necrosis factor (TNF-α), exhibit notable circadian oscillations when they enter the brain, leading to sleep disorders [38, 39]. This relationship between inflammatory mediators and psychiatric symptoms has been previously demonstrated through the link between inflammation and depression [40]. Social isolation also plays a key role in sleep disorders. Sleep disorders were reported to increase during quarantine periods due to COVID-19 infection [41]. During COVID-19 lockdowns, insomnia, poor sleep maintenance, and reduced sleep quality increased for the general population in several countries. This has been attributed to the effect of confinement, worsened by the psychological effects of the disease and increased exposure to artificial lighting from electronic devices [6, 42, 43]. 

References

34. Burks SM, Rosas-Hernandez H, Alejandro Ramirez-Lee M, Cuevas E, Talpos JC. Can SARS-CoV-2 infect the central nervous system via the olfactory bulb or the blood-brain barrier? Brain Behav Immun. 2021;95:7-14.

35. Granholm AC. Long-Term Effects of SARS-CoV-2 in the Brain: Clinical Consequences and Molecular Mechanisms. J Clin Med. 2023;12(9).

37. Hojyo S, Uchida M, Tanaka K, Hasebe R, Tanaka Y, Murakami M, Hirano T. How COVID-19 induces cytokine storm with high mortality. Inflamm Regen. 2020;40:37.

38. Semyachkina-Glushkovskaya O, Mamedova A, Vinnik V, Klimova M, Saranceva E, Ageev V, et al. Brain Mechanisms of COVID-19-Sleep Disorders. Int J Mol Sci. 2021;22(13).

39. Agorastos A, Hauger RL, Barkauskas DA, Moeller-Bertram T, Clopton PL, Haji U, et al. Circadian rhythmicity, variability and correlation of interleukin-6 levels in plasma and cerebrospinal fluid of healthy men. Psychoneuroendocrinology. 2014;44:71-82.

40. Wohleb ES, Franklin T, Iwata M, Duman RS. Integrating neuroimmune systems in the neurobiology of depression. Nat Rev Neurosci. 2016;17(8):497-511. doi:10.1038/nrn.2016.69

41. Pilcher JJ, Dorsey LL, Galloway SM, Erikson DN. Social Isolation and Sleep: Manifestation During COVID-19 Quarantines. Front Psychol. 2021;12:810763.

42. Limongi F, Siviero P, Trevisan C, Noale M, Catalani F, Ceolin C, et al. Changes in sleep quality and sleep disturbances in the general population from before to during the COVID-19 lockdown: A systematic review and meta-analysis. Front Psychiatry. 2023;14:1166815.

43. Karkala A, Tzinas A, Kotoulas S, Zacharias A, Sourla E, Pataka A. Neuropsychiatric Outcomes and Sleep Dysfunction in COVID-19 Patients: Risk Factors and Mechanisms. Neuroimmunomodulation. 2023;30(1):237-49.

4. The authors should clarify the severity of COVID-19 cases included in the study and discuss how this factor might influence the outcomes.

Response: Thank you for the comment. We agree that the severity of COVID-19

---

## [Decision Letter · Decision Letter 1]

20 May 2024

PONE-D-23-38335R1Evaluating the risk of sleep disorders in subjects with a prior COVID-19 infectionPLOS ONE

Dear Dr. Kim,

Thank you for submitting your manuscript to PLOS ONE. After careful consideration, we feel that it has merit but does not fully meet PLOS ONE’s publication criteria as it currently stands. Therefore, we invite you to submit a revised version of the manuscript that addresses the points raised during the review process.

We look forward to receiving your revised manuscript.

Kind regards,

Braja Gopal Patra, Ph.D.

Academic Editor

PLOS ONE

Reviewers' comments:

Reviewer's Responses to Questions

**Comments to the Author**

1. If the authors have adequately addressed your comments raised in a previous round of review and you feel that this manuscript is now acceptable for publication, you may indicate that here to bypass the “Comments to the Author” section, enter your conflict of interest statement in the “Confidential to Editor” section, and submit your "Accept" recommendation.

Reviewer #2: All comments have been addressed

Reviewer #3: (No Response)

2. Is the manuscript technically sound, and do the data support the conclusions?

Reviewer #2: Yes

Reviewer #3: (No Response)

3. Has the statistical analysis been performed appropriately and rigorously? 

Reviewer #2: Yes

Reviewer #3: No

4. Have the authors made all data underlying the findings in their manuscript fully available?

Reviewer #2: Yes

Reviewer #3: No

5. Is the manuscript presented in an intelligible fashion and written in standard English?

Reviewer #2: Yes

Reviewer #3: (No Response)

6. Review Comments to the Author

Reviewer #2: All the comments are well addressed. The manuscript is now in publishable form. Other typos have also been addressed.

Reviewer #3: The authors studied the impact of covid on sleep disorders using the Utah all payers claims database. The results of the study can provide valuable insights to our understanding of covid which can benefit clinicians, public health workers, and patients. The study is well-designed, although I believe the analysis should be improved before the manuscript is ready for publication.

Elixhauser comorbidity index is for predicting in-hospital mortality and resource use. I’d assume most covid or sleep disorders patients would not require hospitalization, so can the authors justify why the included the Eixhauser comorbidity index? Also, some dx included in deriving the Eixhauser comorbidity index overlap with some covariates the authors included in the models (e.g., depression, obesity), so these variables were somehow adjusted twice in the models.

Entropy balancing is for casual inference. Although from the study design I can see the authors wanted to do some causal inference (dropped people with sleep disorders dx before 2021), but then they why did you balance the covariates and ALSO control for them in the logistic models?

I don’t see the point of separate models for private and medicaid beneficiaries (tables 3 and 4). The full model (table 2) already included insurance type (medicaid coverage indicator) as a main effect, which means you were assuming that the effects of all other covariates would be the same for medicaid and private beneficiaries, which seems to be a valid assumption since the two subgroup models (tables 3 and 4) had similar results. If you expect the effects of covariates to vary by insurance type, you should include interaction terms between medicaid coverage and all other covariates in the full model (table 2).

Discussion — It’s very hard to follow some of the literature summaries in the discussion section. E.g, in the second paragraph, the authors started of the paragraph saying their results were consistent with what had been published. But later in the paragraph they mentioned more studies reported “however, sleep disorders … as a long-term effect associated with prior COVID-19 infection”. From the summaries of the studies, It seems to me that the first few studies before the transition sentence (however…) were cross-sectional studies, and studies after the transition sentence were longitudinal studies looking at long term effects. But the starting sentences did not send off this message and were somehow misleading.

Some references were missing — authors mentioned xxx et al. did yyy, but there was no citation at the end. I highly recommend the authors carefully go through the draft again before resubmission. There are some broken sentences.

7. PLOS authors have the option to publish the peer review history of their article (what does this mean?). If published, this will include your full peer review and any attached files.

Reviewer #2: **Yes: **Muskan Garg

Reviewer #3: No

---

## [Author Response · Author response to Decision Letter 1]

3 Jun 2024

May 25, 2024

Dear Dr. Patra and Reviewers, 

We appreciate the opportunity to revise our manuscript, Evaluating the risk of sleep disorders in subjects with a prior COVID-19 infection. Below are the specific reviewer’s comments (in bold) followed by our responses. Newly added parts in the manuscript are highlighted in yellow. Thank you again for taking the time to consider our manuscript. 

Reviewer #2: All the comments are well addressed. The manuscript is now in publishable form. Other typos have also been addressed.

Response: Thank you for taking the time to review the revised manuscript. 

Reviewer #3: The authors studied the impact of covid on sleep disorders using the Utah all payers claims database. The results of the study can provide valuable insights to our understanding of covid which can benefit clinicians, public health workers, and patients. The study is well-designed, although I believe the analysis should be improved before the manuscript is ready for publication.

Elixhauser comorbidity index is for predicting in-hospital mortality and resource use. I’d assume most covid or sleep disorders patients would not require hospitalization, so can the authors justify why the included the Eixhauser comorbidity index? Also, some dx included in deriving the Eixhauser comorbidity index overlap with some covariates the authors included in the models (e.g., depression, obesity), so these variables were somehow adjusted twice in the models.

Response: Thank you for your comment. Studies have indicated that sleep disorders, including sleep apnea and insomnia, may be associated with an increased risk of mortality (Zolfaghari et al., 2024; Cappuccio et al., 2010; Lin et al., 2023). Additionally, the Centers for Disease Control and Prevention reported 848,943 deaths due to COVID-19 in the United States between 2020 and 2021 (source: https://www.cdc.gov/nchs/covid19/mortality-overview.htm). Therefore, we deemed it important to control for a comorbidity index such as Elixhauser.

The Elixhauser comorbidity index assigns weights to each condition and then sums these weights for each subject. Because this index considers multiple diseases dependently, we believe that controlling for individual comorbidities, even though they are included in the Elixhauser index calculation, would be beneficial for enhancing the overall goodness of fit of the regression.

In response to the comment above, we conducted a regression excluding the Elixhauser comorbidity index and only controlled for the individual comorbidities. The results, including odds ratios and p-values of the control variables, remained consistent with and without the inclusion of the Elixhauser index variable. Specifically, the odds ratio of the independent variable (i.e., COVID infection: OR=1.53, p<0.01 in Table 2) remained unchanged in both regressions. Subsequently, we updated the previous results with those from the regression excluding the Elixhauser index variable and made corresponding revisions to the Results section.

References

Zolfaghari S, Keil A, Pelletier A, Postuma RB. Sleep disorders and mortality: A prospective study in the Canadian longitudinal study on aging. Sleep Med. 2024 Feb;114:128-136. doi: 10.1016/j.sleep.2023.12.023

Cappuccio FP, D'Elia L, Strazzullo P, Miller MA. Sleep duration and all-cause mortality: a systematic review and meta-analysis of prospective studies. Sleep. 2010 May;33(5):585-92. doi: 10.1093/sleep/33.5.585.

Lin Y, Wu Y, Lin Q, et al. Objective Sleep Duration and All-Cause Mortality Among People With Obstructive Sleep Apnea. JAMA Netw Open. 2023;6(12):e2346085. doi:10.1001/jamanetworkopen.2023.46085

Centers for Disease Control and Prevention. COVID-19 Mortality Overview. Access on May 22, 2024 at https://www.cdc.gov/nchs/covid19/mortality-overview.htm.

Entropy balancing is for casual inference. Although from the study design I can see the authors wanted to do some causal inference (dropped people with sleep disorders dx before 2021), but then they why did you balance the covariates and ALSO control for them in the logistic models?

Response: Thank you for the comment. Controlling the covariates that were used in the Entropy Balancing enhances the robustness of the estimation through doubly robust estimation. If either the Entropy Balancing model or the main regression model is correctly specified, the estimation remains consistent. This approach offers increased protection against model misspecification and enhances the reliability of causal inference in the study.

I don’t see the point of separate models for private and medicaid beneficiaries (tables 3 and 4). The full model (table 2) already included insurance type (medicaid coverage indicator) as a main effect, which means you were assuming that the effects of all other covariates would be the same for medicaid and private beneficiaries, which seems to be a valid assumption since the two subgroup models (tables 3 and 4) had similar results. If you expect the effects of covariates to vary by insurance type, you should include interaction terms between medicaid coverage and all other covariates in the full model (table 2).

Response: Thank you for the comment. We wanted to explore potential differences in the main effect, specifically the odds ratio of the COVID-19 infection variable, as we considered subgroups based on insurance type (Medicaid only vs. private insurance only). Individuals covered by Medicaid may experience a more pronounced negative impact (OR=1.67, p<0.01) on sleep problems due to COVID-19 infection compared to those covered by private insurance (OR=1.51, p<0.01), although direct comparisons of odds ratios are not feasible (i.e. two regressions with two different number of subjects).

Having interaction terms between Medicaid and all other covariates in the full model has to control over 25 additional variables in Table 2. Also, comparing any differences in Medicaid vs. private insurance is not the main purpose of the paper. Thus, we would like to keep Table 2 as it is. Now, we removed Table 3 (i.e. regression results for those covered by private insurance) and 4 (i.e. regression results of those covered by Medicaid). S2 Table (Association between COVID-19 severity and risk of sleep disorders) has been moved to Table 3. In this model, we removed the Elixhauser index variable and reran the regression. Relevant texts were revised in the paper. 

Discussion — It’s very hard to follow some of the literature summaries in the discussion section. E.g, in the second paragraph, the authors started of the paragraph saying their results were consistent with what had been published. But later in the paragraph they mentioned more studies reported “however, sleep disorders … as a long-term effect associated with prior COVID-19 infection”. From the summaries of the studies, It seems to me that the first few studies before the transition sentence (however…) were cross-sectional studies, and studies after the transition sentence were longitudinal studies looking at long term effects. But the starting sentences did not send off this message and were somehow misleading.

Response: Thank you for the comment. In the paragraph, we first tried to present studies that reported incidences of sleep disorders during the COVID-19 pandemic, followed by studies that reported sleep disorders as a long-term effect associated with prior COVID-19 infection. To address your concerns, we have restructured the flow of the paragraph to make it less confusing and easy to understand. We also changed “however” to “moreover” as the transition word since the studies reported incidences of sleep disorder both during and after COVID infection. 

Some references were missing — authors mentioned xxx et al. did yyy, but there was no citation at the end. I highly recommend the authors carefully go through the draft again before resubmission. There are some broken sentences.

Response: The authors went through the manuscript and the references to make sure. 

Thank you again for taking the time to review the manuscript!

---

## [Decision Letter · Decision Letter 2]

7 Aug 2024

PONE-D-23-38335R2Evaluating the risk of sleep disorders in subjects with a prior COVID-19 infectionPLOS ONE

Dear Dr. Kim,

Thank you for submitting your manuscript to PLOS ONE. After careful consideration, we feel that it has merit but does not fully meet PLOS ONE’s publication criteria as it currently stands. Therefore, we invite you to submit a revised version of the manuscript that addresses the points raised during the review process.

We look forward to receiving your revised manuscript.

Kind regards,

Braja Gopal Patra, Ph.D.

Academic Editor

PLOS ONE

Journal Requirements:

Additional Editor Comments:

Thank you for thoroughly addressing the reviewers' comments. Here are a few additional minor comments from the reviewers that need attention.

Reviewers' comments:

Reviewer's Responses to Questions

**Comments to the Author**

1. If the authors have adequately addressed your comments raised in a previous round of review and you feel that this manuscript is now acceptable for publication, you may indicate that here to bypass the “Comments to the Author” section, enter your conflict of interest statement in the “Confidential to Editor” section, and submit your "Accept" recommendation.

Reviewer #3: All comments have been addressed

Reviewer #4: (No Response)

2. Is the manuscript technically sound, and do the data support the conclusions?

Reviewer #3: Yes

Reviewer #4: Yes

3. Has the statistical analysis been performed appropriately and rigorously? 

Reviewer #3: I Don't Know

Reviewer #4: Yes

4. Have the authors made all data underlying the findings in their manuscript fully available?

Reviewer #3: No

Reviewer #4: No

5. Is the manuscript presented in an intelligible fashion and written in standard English?

Reviewer #3: Yes

Reviewer #4: Yes

6. Review Comments to the Author

Reviewer #3: I appreciate the authors taking time to address all my comments. For the controlling for covariates in regression models after Entrophy Balancing part, I'd recommend the authors mention this in the "statistical approach" section that the weighted regression model also controlled for covariates, and cite a few published papers that have done so.

One other minor comment: P3, line 49, “sleep disorders may be a result of” — there was an additional “as” before “a result of” in the manuscript.

Reviewer #4: The paper studies the impact of covid -19 on sleep disorders.This is done by building a weighted logistic regression model taking patient characteristics including the occurrence of COVID-19 as independent variables and occurrence of sleep disorders as dependent variable.

Pros

1. The paper is detailed, well written and easy to understand.

2. the authors sufficiently addressed reviewer concerns

Cons

1. The AUC/ROC for logistic regression is not included

2. There is no description of the weighting scheme used by the logistic regression model.

7. PLOS authors have the option to publish the peer review history of their article (what does this mean?). If published, this will include your full peer review and any attached files.

Reviewer #3: No

Reviewer #4: No

---

## [Author Response · Author response to Decision Letter 2]

15 Sep 2024

September 03, 2024

Dear Dr. Patra and Reviewers, 

We appreciate the opportunity to revise our manuscript, Evaluating the risk of sleep disorders in subjects with a prior COVID-19 infection. Below are the specific reviewer’s comments (in bold) followed by our responses. Newly added parts in the manuscript are highlighted in yellow. Thank you again for taking the time to consider our manuscript. 

Reviewer #3: : I appreciate the authors taking time to address all my comments. For the controlling for covariates in regression models after Entrophy Balancing part, I'd recommend the authors mention this in the "statistical approach" section that the weighted regression model also controlled for covariates, and cite a few published papers that have done so.

Response: Thank you for the comment. We added the following sentence in the Statistical Approach section with two references:

The weighted regression model controlled for the covariates that were included in the EB.

References 

Sudduth, J. D., et al. Preoperative opioid use and its association with postoperative complications. Journal of Substance Use. 2024: 1–7. https://doi.org/10.1080/14659891.2024.2351016. 

Ricci C., et al. Appendiceal goblet cell carcinoma has marginal advantages from perioperative chemotherapy: a population-based study with an entropy balancing analysis. Langenbecks Arch Surg. 2023 Jan 25;408(1):65. doi: 10.1007/s00423-023-02791-x. 

 One other minor comment: P3, line 49, “sleep disorders may be a result of” — there was an additional “as” before “a result of” in the manuscript.

Response: Thank you for pointing it out. It was an error, and we have corrected it as follows:

“Sleep disorders may be a result of various factors ranging from stressful life events…”

Reviewer #4: The paper studies the impact of covid -19 on sleep disorders. This is done by building a weighted logistic regression model taking patient characteristics including the occurrence of COVID-19 as independent variables and occurrence of sleep disorders as dependent variable.

1. The AUC/ROC for logistic regression is not included. 

Response: Thank you for the comment. The area under the ROC curve (AUC) for both Tables 2 and 3 is 0.72, indicating that there is a 72% chance that the model is able to distinguish between positive cases and negative cases. 

ROC for Table 2:

ROC for Table 3:

According to Hosmer & Lemeshow (Applied logistic regression, 2013), our AUC/ROC=0.72 offers acceptable discrimination. 

0.5 ≤ no discrimination

0.5-0.7 = poor discrimination

0.7-0.8 = Acceptable discrimination

0.8-0.9= Excellent discrimination

>0.9 = Outstanding discrimination

We added the following sentences in the Statistical Approach section:

The area under the receiver operating characteristic (ROC) curve (AUC) was used to measure the performance of the logistic regression models. 

We added the following result in the Results section:

The AUC following the logistic regressions were 0.72 for both Tables 2 and 3, indicating that there is a 72% chance of the models distinguishing between positive and negative cases. 

Reference 

Hosmer Jr., D.W., Lemeshow, S. and Sturdivant, R.X. (2013) Applied Logistic Regression. 3rd Edition, John Wiley & Sons, Hoboken, NJ.

2. There is no description of the weighting scheme used by the logistic regression model.

Response: Thank you for the comment. We added the following parts in the Statistical Approach section: 

The EB approach for a binary outcome is based on a maximum entropy reweighting scheme that assigns weights to each subject so that the control group (i.e. no COVID infection group in this study) is reweighted to match the controlled covariate moments (i.e. mean, standard deviation, and variance in this study) in the treatment group (i.e. COVID infection group in this study). Therefore, EB ensures that the two groups being compared are similar enough by reweighting the patient characteristics (i.e. covariates)

We have added three references for these parts:

Zhao, Qingyuan and Percival, Daniel. "Entropy Balancing is Doubly Robust" Journal of Causal Inference, vol. 5, no. 1, 2017, pp. 20160010. https://doi.org/10.1515/jci-2016-0010

Hainmueller, J., & Xu, Y. (2013). ebalance: A Stata Package for Entropy Balancing. Journal of Statistical Software, 54(7), 1–18. https://doi.org/10.18637/jss.v054.i07

Markoulidakis, A., Taiyari, K., Holmans, P. et al. A tutorial comparing different covariate balancing methods with an application evaluating the causal effects of substance use treatment programs for adolescents. Health Serv Outcomes Res Method 23, 115–148 (2023). https://doi.org/10.1007/s10742-022-00280-0

We appreciate your time to review the manuscript!

---

## [Editor Report · Decision Letter 3]

27 Sep 2024

Evaluating the risk of sleep disorders in subjects with a prior COVID-19 infection

PONE-D-23-38335R3

Dear Dr. Kim,

We’re pleased to inform you that your manuscript has been judged scientifically suitable for publication and will be formally accepted for publication once it meets all outstanding technical requirements.

Kind regards,

Braja Gopal Patra, Ph.D.

Academic Editor

PLOS ONE

Additional Editor Comments (optional):

The authors have addressed all reviewer comments.
---

## [Editor Report · Acceptance letter]

7 Oct 2024

PONE-D-23-38335R3 

PLOS ONE

Dear Dr. Kim, 

I'm pleased to inform you that your manuscript has been deemed suitable for publication in PLOS ONE. Congratulations! Your manuscript is now being handed over to our production team.

Kind regards, 

on behalf of

Dr. Braja Gopal Patra 

Academic Editor

PLOS ONE